# Hierarchical Polyaniline Core-Shell Nanocomposites Coated on Modified Graphite for Improved Electrical Conductivity Performance

**DOI:** 10.3390/nano12213776

**Published:** 2022-10-26

**Authors:** Asima Naz, Ali Irfan, Sami A. Al-Hussain, Iram Nawaz, Shah Faisal, Rabia Sattar, Magdi E. A. Zaki

**Affiliations:** 1Department of Chemistry, Mirpur University of Science & Technology (MUST), Mirpur Azad Jammu & Kashmir 10250, Pakistan; 2Department of Polymer Science & Engineering, Zhejiang University, Hangzhou 310027, China; 3Department of Chemistry, Government College University Faisalabad, Faisalabad 38000, Pakistan; 4Department of Chemistry, College of Science, Imam Mohammad Ibn Saud Islamic University (IMSIU), Riyadh 13623, Saudi Arabia; 5Abdul Razzaq Fazaia College, M.M. Alam Base Mianwali, Mainwali 42206, Pakistan; 6Department of Chemistry, Islamia College University Peshawar, Peshawar 25120, Pakistan; 7Department of Chemistry, The University of Lahore, Sargodha Campus, Sargodha 40100, Pakistan

**Keywords:** multilayered nanocomposites, conducting filler, in situ polymerization, physical properties, electrical conductivity

## Abstract

Graphite has recently gained scientific and industrial attention due to its high electrical conductivity. In the current endeavor, a new way to fabricate novel and multifunctional nanocomposites using functional graphite (FG) as filler is presented. The fabrication of multilayered conducting composites of PANi/PMMA/PPG-b-PEG-b-PPG was carried out via in situ polymerization, using polyaniline (PANi), poly(methyl methacrylate) (PMMA) and block copolymer as matrices in the presence of FGfiller. The growth of PANi chains is manifested by PMMA due to the formation of H-bonding between imine and carbonyl groups of PANi and MMA units, respectively, and are responsible for ion exchange sites. FTIR spectroscopy was used for structural elucidation of composites while elemental analysis was accomplished by XPS and EDX spectroscopy. The morphology of the prepared PANi/PMMA/PPG-b-PEG-b-PPG@FG composites was inspected by the SEM. The structure and crystallinity of the composites was investigated via XRD. The improved thermal stability and properties of the nanocomposites were observed using TGA and DSC. The conductivity measurements were used to characterize the electrical conductivity performance of the resulting composites. The presence of functional filler as well as polyaniline shows a significant contribution towards the enhancement of electrical conductivity of PANi/PMMA/PPG-b-PEG-b-PPG@FG nanocomposites.

## 1. Introduction

In recent eras, polymer nanocomposites reinforced even with a small amount of filler (5%) have gained much interest owing to their improved thermomechanical characteristics including increased strength and elastic modulus [1]. In particular, electrically conducting polymer (CP) nanocomposites have been the main focus of study for researchers because of their usage in batteries, light-emitting (LE) devices, electromagnetic shielding (EMS), semiconductors, corrosion resistance, anti-static coatings and some other practical applications [2]. However, incompatibility of fillers with polymers can lead to poor dispersion and agglomeration due to their large surface areas. Thus, noteworthy endeavor shave been directed to enhance the electrical conductivity performance of the composites by the incorporation of conducting fillers. The commonly used fillers are carbon black (CB), metal powders, carbon fibers (CFs) and graphite(G). The reported electrical conductivities for polymers are 10^−14^ to 10^−17^ S/cm, 10^5^ for high-purity synthetic graphite, 10^6^ for metals including copper and aluminum, 10^2^ for carbon black and 10^4^ for natural abundant graphite [3]. Such higher conductivity values of carbon-based fillers such as graphene, carbon nanotubes and carbon fibers make them auspicious fillers for conductive composites, but these materials are currently too costly to apply on an industrial scale [4,5,6,7,8,9,10]. Therefore, expanded graphite (EG) obtained from graphite intercalation compounds being exposed to microwave radiations or by rapid heating may be alternative filler.EG can be produced massively with a comparatively low cost and relatively higher conductivity value [2]. Wei et al. [11] reported the preparation of EG using microwave irradiation methodover4 min, consisting of 3 min mixing and 1 min of microwave irradiation. Yang et al. [12] carried out the fabrication of acrylonitrile butadiene rubber (NBR)/EG composites via latex compounding method and observed that thermal conductivity for unfilled NBR composites was improved from 0.19 Wm^−1^K^−1^ to 0.30 Wm^−1^K^−1^ for NBR/EG composites. Mu and Feng [13] studied the influence of preparation method on thermal conductivity of silicone rubber (SR)/EG composites and it was found that relative to melt mixing, the most effective method is solution intercalation in promoting the thermal transport of SR/EG composites.

There are numerous investigations focusing on exfoliated graphite composites with a variety of polymers such as polystyrene (PS) [14,15,16], low-density polyethylene (LDPE) [3], polyaniline (PANi) [5] and polypyrrole (PPy)/copolymers [6,17,18]. In all aforementioned studies, EG significantly enhanced the electrical conductance performances of the prepared blends/nanocomposites. In addition to EG, the use of CP could also improve the electrical and thermal executions of the nanocomposites. However, the poor mechanical properties of CP and its inability to be processed by conventional methods are two major limitations for the use of conducting polymer. These limitations can be overcome by the preparation of CP blends and composites possessing electrical properties of conducting polymer and mechanical exploits of host matrix [19]. The use of a compatibilizer including copolymer is another method to improve the compatibility in polymer blends [20,21,22,23]. The mechanical behavior of polymer nanocomposites can also be affected largely by the presence of graphite particles. For polymer composites, graphite platelets are a class of well-known additives, but stacking of graphene layers is also another dispiriting challenge in polymer matrices. However, the functionalization process of graphite and a suitable fabrication approach can formulate the high-performance composites with improved mechanical, thermal and electrical properties.

In this study, multilayered PANi/PMMA/PPG-b-PEG-b-PPG nanocomposites were prepared via in situ polymerization using functionalized graphite (FG) filler to examine the effect of FG on various characteristics of multilayered composites such as structure, morphology, and thermal and electrical behavior. Moreover, the influence of conducting polymer (PANi) was also studied on synthesized nanocomposites by changing its concentration while keeping filler concentration constant, as only a little study on the fabrication of multilayered nanocomposites using PANi/FG as a core has been attempted to date. Fourier transform infrared spectroscopy (FT-IR), X-ray photon spectroscopy (XPS), energy dispersive X-ray spectroscopy (EDX) and X-ray diffraction were used to analyze the chemical composition whereas the morphology of PANi/PMMA/PPG-b-PEG-b-PPG@FG composites was investigated by scanning electron microscopy. Thermal characteristics of finally achieved composites were examined by thermogravimetric analysis (TGA) and differential scanning calorimetry (DSC). Lastly, the synergetic conductivity effect of PANi accompanying FG was also studied in addition to thermal stability.

## 2. Materials and Methods

### 2.1. Materials

Polyaniline (PANi, 99%), Polymethyl methacrylate (PMMA, M_w_~120,000), Poly (propylene glycol)-block-Poly(ethylene glycol)-block-poly(propylene glycol) (PPG-b-PEG-b-PPG, M_n_~4400) and fine graphite powder (99%) were purchased from Sigma-Aldrich (St Louis, MO, US). Other reagents such as nitric acid (HNO_3_, 70%), hydrochloric acid (HCl, 37%), potassium dichromate (K_2_Cr_2_O_7_, 99.99%) and dichloromethane (DCM) were also procured from Sigma-Aldrich and used as received.

### 2.2. Sample Preparation

#### 2.2.1. Purification and Functionalization of Graphite

Prior to functionalization, graphite was first purified to remove the various metal contents and other carbonaceous impurities such as fullerene incorporated during its synthesis. Purification was performed by refluxing the required amount of graphite with 50 mL HCl at 65 °C for 2 h followed by filtration and washing with distilled water to attain pH = 7 by removing excess of HCl and dried at 90 °C. After purification, functionalization of graphite was performed. For this drive, required quantities of purified graphite and HNO_3_ were mixed at ambient temperature and refluxed for 24 h at 140 °C with constant stirring. Afterwards, 500 mL of deionized water (DI) was added to the above mixture, filtered and washed several times with DI to attain neutral pH [24]. The functional graphite (FG) was finally dried for 2 h at 60 °C.

#### 2.2.2. Preparation of Multilayered PANi/PMMA/PPG-b-PEG-b-PPG@FG Nanocomposite

For the preparation of multilayered nanocomposites, 2 mL of aniline was added to 200 mL of HCl (1M, aqueous) and stirred for 1 h. Later on, FG (5 g) was also added to above mixture and followed by very slow addition of 200 mL of K_2_Cr_2_O_7_/HCl (1M) solution. The temperature of the reaction mixture was kept at 0 °C under continuous stirring to initiate the oxidation process. The resultant mixture was vacuum filtered using 0.4 mm pore size filter paper and dried at 60 °C for 2 h (labelled as 1A). Subsequently, the 4 g of 1A in 300 mL of DCM was poured into a two-necked round-bottom flask and 4 g of PMMA/DCM (100 mL)was also added to it. The entire mixture was refluxed for 8 h at 50 °C and then cooled to room temperature followed by filtration and drying at 60 °C (labelled as 1B). The multilayered nanocomposite was fabricated by the addition of a solution containing 3 g of PPG-b-PEG-b-PPG/DCM (30 mL) into 3 g of 1B/DCM (250 mL) dispersion. The whole reaction mixture was refluxed for 12 h at 50 °C and final filtered product Composite-I (C1) was dried at 60 °C. Similarly, different series were prepared by changing the concentrations of conducting polymer as listed in Table 1. The formation of multilayered composites is illustrated in Figure 1.

### 2.3. Characterization

Nexus 870 FTIR spectrometer (Wisconsin, USA) was used to record the FTIR spectra of prepared samples in transmittance mode in the range of 4000–500 cm^−1^. XPS analysis was performed to confirm the surface composition of the samples using a Perkin-Elmer PHI system. An EDX-720/800HS/900HS (Shimadzu Europe, Germany) spectrometer was used for elemental analysis. Field emission SEM of the fractured samples under nitrogen was accomplished using a Hitachi S-4800 (Tokyo, Japan). TGA/DTA (Mettler Toledo, FL, USA) thermogravimetric analyzer was used to investigate the thermal behavior of the fabricated composites. A total 5 mg of the sample in an Al_2_O_3_ crucible was used at a heating rate of 10 °C min^−1^. A Mettler Toledo DSC 822 (Switzerland) was used to inspect the phase change temperatures of the samples (5–10 mg) in aluminum pans at a heat rate of 10 °C min^−1^. PW 3040/60 X’pert PRO (PANalytical, Netherland) X-ray diffractometer was employed for the XRD patterns using Ni-filtered Cu Kα (40 kV, 30 mA) radiations. Micromeritics Trisstar II automatic adsorption instrument was used to evaluate the surface area of the prepared samples by N_2_ adsorption–desorption isotherms (77 K). Keithley 2401 (Cleveland, OH, USA) conductivity meter was used to measure the electrical conductivity of 25 × 10 × 1 ± 0.05 mm^3^ thin films of the multilayered composites under ambient atmosphere using four-probe method.

## 3. Results and Discussion

### 3.1. FTIR Spectroscopy

The tabulated FTIR data (Appendix A) establish the structure of FG and different polymer metrices used in this study deposited over FG. The assigned infrared spectra for FG, PANi@FG, PANi/PMMA@FG and PANi/PMMA/PPG-b-PEG-b-PPG@FG nanocomposites are displayed in Figure 1. It can be clearly seen that PANi@FG exhibits more peaks compared with functional graphite, which are representative of PANi@FG (Figure 1b). Two characteristic peaks for N-H stretch and electronic band of PANi are located at 3425 cm^−1^ and 1118 cm^−1^, respectively. The band at 1567 cm^−1^ is ascribed to benzenoid vibrations in PANi, and the appearance of peaks at 1728 cm^−1^ and 1055–1412 cm^−1^ correspond to C=O in COOH and C-O in COH functional groups that were introduced on the surface of graphite by functionalization of graphite. Compared with FG, the C=O peak shifted downward, i.e., 1721 cm^−1^, in the case of PANi@FG. Similarly, the peak located at 1561 cm^−1^ for functionalized graphite also shows red shift with increased intensity, showing the H-bonded carboxyl groups. The hydrogen bonding and π–π interaction between sheets of graphite and polymer backbone are responsible for this red shift [1,2]. The absorption band positioned at 1293 cm^−1^ is characteristic of the protonated form of PANi. The band at 1475 cm^−1^ is attributable to C=C stretching mode for both quinoid and benzenoid rings. However, the incorporation of FG particles leads to shift of some bands of PANi. Two characteristic peaks of PANi are present in composite at 3438 cm^−1^ and 1167 cm^−1^ and are representative of N-H stretch and an electronic-like band [3,4,21,22]. The band appearing at 1538 and 1629 cm^−1^ corresponds to benzenoid and quinoid vibrations in PANi, respectively. The band corresponding to C-N stretching vibrations in the polaron structure appears at 1398 cm^−1^ whereas C-H out-of-plane bending vibrations are represented by the appearance of the peak at 795 cm^−1^. The band appearing at 1167 cm^−1^ is associated with high electrical conductivity. It shows a high degree of electronic delocalization in synthesized PANi@FG. The two peaks of almost equal intensity at 1440 cm^−1^ and 1475 cm^−1^ demonstrate the deposition of PANi in emeraldine state [5]. The characteristic absorption bands of PMMA can be clearly seen from Figure 1c, such as the peaks at υ = 2921.75, 2967.18 and 1290.50 (–CH3), and at 1731.85 (C=O) cm^−1^. The region at υ= 3000–3192 was attributed to having no monomer residual in the composite. The band was assigned to carbonyl group (C=O) of PMMA at 1731 cm^−1^ in the blend of PANi and PMMA. The other small peaks at around 1117 cm^−1^ are owing to stretching vibrations of C-O, which are present in ester group of PMMA. Besides this, the peaks for quinoid ring of PANi, which appeared at 1475 cm^−1^ in PANi, now appear at 1482 cm^−1^ because of the slight shifting of peak towards the high wave number side with reduced intensity corresponding to NH bending vibration mode in PANi. These changes represent the formation of H-bonding between characteristic groups of PANi and PMMA. H donation in the NH group allowed the compatibility of PANi and PMMA, which enhanced the formation of interpenetrating network of PANi and other matrix chains. The splitting of C-O band into a number of other sharp peaks shows the stretching vibrations of PANi in C-O absorption peak of PMMA. The broad peak at 3432 cm^−1^ is due to OH group at water molecule adsorbed on matrix PMMA polymer. The rest of the peaks were assigned to the presence of PPG-b-PEG-b-PPG (block copolymer) structures in backbone of polymer composite. In the spectrum of block copolymer, the major absorption peak appears at 2898.67 cm^−1^ for –CH2 asymmetric (C-H) stretching vibration and the peak at 1117 cm^−1^ corresponds to C-O stretching vibration [16,23,24]. The decrease in peak intensities also shows the deposition of block copolymer in the form of clathrates, as represented in Figure 1d.

### 3.2. X-ray Photoelectron Spectroscopy

The surface features of multilayered PANi/PMMA/PPG-b-PEG-b-PPG@FG nanocomposites were evaluated in order to confirm the content and binding configurations of elements present. The XPS spectrum of PANi@FG composites shows the existence of primary C1s, O1s and N1s core level (Figure 2a–c). The atomic percentages of carbon, hydrogen and nitrogen are 79.65, 7.87 and 12. 47%, respectively. The O1s peak arises mainly from some oxygen or water absorbed onto the surface while nitrogen peak appeared due to the coated PANi. XPS of C1s ranging from 280–300 eV is convoluted into three peaks centered at 284.5, 285.5 and 286.5 eV by curve fitting of C1s spectrum of PANi@FG. In addition, another peak (N1s) on XPS spectrum of PANi@FG nanocomposites was noticed, which was referenced to nitrogen atoms of PANi indicating PANi covering on the surface of FG. The N1s spectrum can be deconvoluted into three peaks located at 399.0, 401.0 and 400.0 eV. The peak at 401.0 is correlated with protonated amine unites atoms while peak at 400.9 is associated with cationic nitrogen atoms (polaron and bipolaron). The shift of peak towards higher binding energy is owing to electron localization associated with poor conjugation at SP3-bonded sites. As evidenced by nitrogen peaks, N atoms are successfully inserted into graphite layers causing a large number of topological defects and, hence, forming a disordered layered structure, which can ultimately facilitate Li-ion intercalation. These electronic defects were produced in polymer chains by the protonation of PANi: the formation of polaron and bipolaronic takes place by the addition of protons to neutral polymer chains. Kumar et al. [25] investigated whether these defects are related to two charged nitrogen species. Figure 2d shows the C 1s spectrum for PANi/PMMA@FG nanocomposites. This C 1s XPS spectrum shows four types of carbon: carbon that is singly bonded to carboxyl (C-CO2) appears at 285.0 eV; the peak at 284.6 eV reflects carbon bonded to hydrogen; peak at 286.1 eV attributes to -C-O-C-and appearance of peak at 288.3 eV indicates the existence of carbon of carboxyl. Based on energy transition and energy level structure analyses, it is concluded that the absorption of photon takes place by π-π* transition of C=O group and transference of energy occurs by energy level transition from C=O to C-C and C-O groups. As a result, C-C and C-O bonds breakdown from main and branch chains of polymer molecules [17,18,22,26,27]. It was noticed from the C 1s spectrum of PANi/PMMA/PPG-b-PEG-b-PPG@FG that, in addition to the increase in peak intensity of C-N and O-C=O, there was also disappearance of the C-O peak at 288.3 eV indicating a grafting reaction of PPG-b-PEG-b-PPG on PMMA. The shift of NH and N^+^ peaks towards higher binding energy provide further clear evidence of a successful reaction. As indicated by XPS results of PANI/PMMA/PPG-b-PEG-b-PPG@FG, C-O and C=O groups are present in composite that strongly interact with other polar groups accommodated by other polymeric chains. So, these carboxyl groups or radicals act as initiators and are responsible for the formation of self-cross-linked polymeric chains over the surface of FG. The corresponding binding energies of respective peaks are disclosed in Table 2.

### 3.3. Energy Dispersive X-ray Spectroscopy (EDX) Analysis

Through EDX analysis, the composition of multilayered nanocomposites PANi/PMMA/PPG-b-PEG-b-PPG@FG was analyzed. Figure 3 presents the EDX spectrum of the prepared nanocomposite comprising of carbon, nitrogen, oxygen, aluminum, silicon, platinum, silver and potassium. The percentage composition values observed are fully related to the type of monomer and catalyst used in polymerization. Results presented confirm the presence of carbon associated with the existence of polymer backbone while the significant amount of nitrogen was attributed to better layering of polymer, i.e., polyaniline (PANi) and oxygen arises as a result of functionalization. Elemental analysis of core shell PANi/PMMA/PPG-b-PEG-b-PPG@FG nanocomposites revealed that the composite was free of impurities but traces of chlorine, aluminum, platinum and potassium were attributed to the catalyst involved in synthetic procedure. PANi/PMMA/PPG-b-PEG-b-PPG@FG elemental analysis showed C = 73.93%, N = 11.13%, O = 10.71%, Al = 1.11%, Si = 1.14%, Pt = 0.57%, Cl = 1.21% and K = 0.19%. Some traces of chlorine impurities were also observed in the nanocomposites owing to acid (HCl) treatment involved in functionalization procedure and can be rid by extensive washing with DI water. These impurities can be removed by extensive washing with deionized water.

### 3.4. Scanning Electron Microscopy

The maximum number of pores can be generated by functionalization without altering the integrity of graphite leading to a significant dispersion of graphite compared with nonfunctional ones. It can be clearly seen from Figure 4a,b that FG appear in the form of large aggregates and have a flaky shape, which was due to presence of micro-sized and nano-sized granules. The graphite sheets were entangled to each other, resulting in big pores in them. The existence of these pores promotes the adsorption of molecular chains as well as monomers onto the pores. The prevalent pores and large surface area assists the processing and development of multilayered nanocomposites. Moreover, as shown, this tangled structure was ripped into smaller pieces; the figure also shows the fractured surface of graphite after functionalization, demonstrating that individual graphite nanosheet is not a single graphite nanosheet or graphene but instead constitutes several layers of graphite sheets. The graphite worms were completely torn into sheets of nanoscale thickness. The composites showed a uniform distribution of graphite particles in PANi matrix demonstrating coating of PANi. Compared with the fragile structure of FG, the composite exhibited a typical structure with the crumple and scroll structure of graphite constituting the chain structure of polyaniline (Figure 4c,d). The morphologies of the composites seem to be different from each other;although, there was a significant change in topography of the samples changing the size of conducting cluster formed by PANi and graphite particles. Figure 4e,f present the SEM micrographs of PANi/PMMA@FG recorded for the fabricated composites. It can be seen from SEM images that crystallites are well packed, constituting better connectivity between crystallites. These are distributed evenly having negligible agglomeration, which is associated with minimum porosity of composites. The well dispersion of FGin polymer matricesis observed as bright spikes with no agglomerate formation in multilayered composites, as shown in Figure 4g,h. This outcome was achieved through the higher degree of interaction between filler and matrix by chemical modification of graphite.The presence of aggregation of the nanofiller in the polymer matrix can disparage the thermal belongings of the multilayered composites [28]. A closer look at the micrographs explores interconnected polymer fiber containing web-like morphology, and such coating of polymers on functional filler is attributed to the high specific surface area as well as the smaller size of graphite providing better adsorption sites for monomers facilitating polymerization. Moreover, some functional group such as –COOH and –OH existed on the surface and pores of the functionalized graphite after having acid treatment as well as temperature treatment promoting the adsorption of monomers onto pores and of molecular chains. The overall conductivity of composites containing bulk material is influenced by the expansion of graphite after acid and thermal treatment rendering graphite flakes as conductive fillers [22,23,24,29].

### 3.5. Thermal Analysis

#### 3.5.1. Thermogravimetric Analysis

The thermal stability of prepared multilayered nanocomposites was studied byTGA. The TGA curves for PANi@FG, PANi/PMMA@FG and PANi/PMMA/PPG-b-PEG-b-PPG@FG nanocomposites are shown in Figure 5, with respective data being summarized in Appendix A. From the TGA curves, it is observed that PANi@FG composite is stable up to 250 °C while weight loss after 390 °C corresponds to complete polymer degradation. The TGA curves are shown following three-step degradation at various temperatures.
(1)First stage starting at 100 °C is considered as initial dehydrating stage corresponding to desorption of water moieties at the surface of polymer.(2)Second stage at 350 °C corresponds to elimination of protonic acid component.(3)Third stage decomposition at 400 °C is related to breakdown of polymer chain leading to production of gases.

The weight loss in the range of 250–350 °C is associated with the exclusion of oxygen containing functional groups, where 88% residue weight indicated retention of some functional groups. Incorporation of PMMA increases the thermal stability of composites. Although no more difference was observed in the degradation temperature (T_20_) of PANi@FG (351 °C) and PANi/PMMA@FG (353 °C), degradation of PANi/PMMA@FG composite occurred at a relatively higher temperature, as shown in Figure 5a,b. Further, residual weight values of PANi/PMMA@FG composite are found to be higher than PANi@FG. In the case of PANi/PMMA/PPG-b-PEG-b-PPG@FG composite (Figure 5c), although a low T_20_ value is observed, the maximum degradation temperature (T_max_) was found to be significantly higher than the PANi@FG and PANi/PMMA@FG, indicating the higher thermal stability of the multilayered composite. The decrease in the initial degradation temperature is due to the moisture content and volatile moieties associated with block copolymer. The PANi/PMMA/PPG-b-PEG-b-PPG@FG composite showed 71% weight retention at 550 °C, which is attributed to carbon net structure in the composite [30,31].

#### 3.5.2. Differential Scanning Calorimetry

The DSC heating traces for PANi@FG, PANi/PMMA@FG and PANi/PMMA/PPG-b-PEG-b-PPG@FG nanocomposites are displayed in Figure 6. The tabulated data (Appendix A) summarize the glass transition temperature (T_g_), melting temperature (T_m_) and crystallization temperature (T_c_), heat of melting (ΔH_m_), and heat of crystallization (ΔH_c_) obtained from DSC. It was noticed that both melt and crystallization temperature increase with the incorporation of polymer matrices. T_c_ for PANi@FG composite appeared at around 172.81 °C, and only 5 °C risewas observed for PANi/PMMA/PPG-b-PEG-b-PPG@FG composite. However, a significant shift of ~18 °C was observed in the melt behavior (T_m_) of the multilayered composite as compared with PANi@FG. Apart from melt behavior, the glass transition temperature (T_g_) observed for polymer matrices in the multilayered composite showed a remarkable shift of ~36 °C compared with PANi@FG. These significant improvements in T_m_ and T_g_ are mainly ascribed to PMMA’s rigid structure. The carbonyl groups of PMMA can easily form H-bonding with the imine groups PANi@FG and are responsible for improved T_g_ and T_m_ as well. Compared with PANi@FG, the final composite also showed increased melt as well as crystallization enthalpies. In addition, well dispersion of modified graphite may also hinder the molecular motion of polymeric chains (by increasing the chain rigidity), which are responsible for an increased T_g_ of multilayered nanocomposites. Appendix A demonstrates the influence of concentration of CPon T_g_ of the prepared multilayered nanocomposites. The glass transition temperature for C1, C2 and C3 was observed at 114, 124 and 120 °C, respectively. Such increased T_g_ values for multilayered composites were ascribed to higher concentrations of conducting polymer, which led to enhanced chain rigidity of polymeric matrices by the formation of H-bonding.

### 3.6. X-ray Diffraction Analysis

The crystalline behavior of the PANI@FG, PANI/PMMA@FG and fabricated multilayered composites I, II and III was investigated by XRD studies. The interlayer distance was predicted using Bragg’s law. The 2θ scan in the range of 10–70° showed the XRD pattern of PANi/PMMA/PPG-b-PEG-b-PPG@FG nanocomposites, as presented in Figure 7. The diffractogram of multilayered PANi@FG, PANi/PMMA@FG and PANi/PMMA/PPG-b-PEG-b-PPG@FG nanocomposites indicated a diffraction peak at 2θ = 27, 26.7 and 26.5°, respectively, while diffraction peaks for composites with change of concentration were visible around 2θ = 26.8, 26.5 and 26.1° accordingly. The peak located at 26 is related to the 002 plane of graphite [32]. The appearance of the characteristic sharp peak at around 2θ = 25–27 depicts that the synthetic procedure was able to successfully insert the functionalized graphite particulates in various polymer networks without affecting the growth of polymer, which still retains its crystalline structure. Noticeably, it is worth mentioning that successful deposition of polymers is clearly evidenced by the shift of diffraction peaks towards lower angle. The typical d-spacing is presented in Appendix A. The tabulated data indicate the slight shift of peak with decreased peak intensity, which is attributed to increased interlayer distance owing to polymerization leading to expansion of graphite layers. In addition, this shift of peak towards lower angle adds to the evidence of breakage of regular graphitic structure constituting stacked layers, which is also consistent with SEM images. The two crystal planes of PANi are reflected by the appearance of two peaks centered at 2θ = 20 and 25°, which are associated with periodicity parallel and perpendicular to polymer chains [10,16,24].

### 3.7. Surface Area and Porous Texture

The adsorption–desorption isotherms for PANi@FG, PANi/PMMA@FG and PANi/PMMA/PPG-b-PEG-b-PPG@FG composites are shown in Figure 8A to investigate their porous structure. Barrett–Joyner–Halenda (BJH) method was employed to calculate the pore size distribution (D_p_) of isotherms. The Brunauer–Emmett–Teller (BET) specific surface area and porosity characteristics of fabricated multilayered nanocomposites are displayed in Appendix A. The isotherms of all samples demonstrated a series of typical adsorption behavior including monolayer adsorption, multilayer adsorption, micropore filling and capillary condensation. It can be observed that most of the adsorptions occur at relatively low pressure with a plateau at a high relative pressure, which indicated a typical microporous structure with very few mesopores. At higher pressure, the final composites PANi/PMMA/PPG-b-PEG-b-PPG@FG showed better adsorption capacity compared with a single-layered PANi@FG nanocomposite whose adsorption capacity is less than 20 cm^3^/g. The surface areas of PANi@FG, PANi/PMMA@FG and PANi/PMMA/PPG-b-PEG-b-PPG@FG composites are 0.554, 1.855 and 4.578 m^2^g^−1^, respectively. The larger surface area provided by the synthesized nanocomposites is useful for the access of electrolytes providing more active sites for the insertion of lithium ions [33]. Figure 8B presents D_p_ derived from adsorption isotherms using BJH model. The enhanced pore volume in the case of composites PANi/PMMA/PPG-b-PEG-b-PPG@FG compared with PAN@FG provides more surface area for adsorption of substrate. The D_p_ = 2.267 nm for multilayered nanocomposites is beneficial for the diffusion of lithium ions (Li^+^) into the active sites [34].

### 3.8. Electrical Conductivity

The electrical conductivity of the core-shell composites, i.e., PANi@FG, PANi/PMMA@FG and PANi/PMMA/PPG-b-PEG-b-PPG@FG was examined by four probe conductivity meters. Figure 9 shows the temperature dependency of conductivity of composites at various temperatures. Initially, the rigid conduction or valence bands do not fill by positive and negative charges and mobility of free charges between polymer chains in conducting polymers are restricted at lower temperatures. However, electrical conductivity increases with an increase in temperature (20–200 °C), as represented in Figure 9, which might be attributed to the increase in free charges with temperature. The multilayered PANi/PMMA/PPG-b-PEG-b-PPG@FG nanocomposites displayed the highest electrical conductivity values (~7 S cm^−1^) compared with PANi/PMMA@FG composite (6.7 S cm^−1^), which, in turn, is higher than PANi@FG composite (~6 S cm^−1^) at the same FG loading fractions. The presence of PANi and PMMA moieties in the multilayered PANi/PMMA/PPG-b-PEG-b-PPG@FG composites provides the percolative conducting bridges among the polymer matrices and FG owing to enhanced electrical conductivity of composites even at lower loadings of filler. Appendix A discloses the electrical conductivity of composites constituting various CP concentrations as a function of temperature. The composites with lower polymer filling showed lower conductivity in comparison with those with higher loadings; this sharp increase, which is also known as percolation transition, emerges when polymer filling reaches a critical value. After attaining this critical value, the conductivity value levels off. The composites with higher CP loadings showed higher conductivity attributed to the enhanced number of conductive paths in graphite-based composites. This transition can be explained by the formation of a conducting network within the framework of multilayered composites. Dispersion of filler is another important factor that plays an important role in variation of percolation threshold for conductivity transition in polymer composites. In our work, we used ultrasonic bath and intensive stirring for a long time to promote fine dispersion of filler into polymer matrices. The sufficient adsorption of PANi and PMMA chains on various pores of graphite was achieved due to good dispersion of graphite flakes into polymers and is responsible for the improved electrical behavior of the fabricated composites. However, at higher concentrations, a slight decrease in conductivity was observed. The reason is that higher concentration of CP may hinder the polymer chains to enter into minor pores due to higher viscosity and, hence, leading to poor dispersion of graphite flakes in polymers. Generally, at higher concentrations of PANi, increased electrical conductivity values may be due to π-π stacking between FG and PANi chains, for which free charge mobility within the composite system increases [21,35,36].

## 4. Conclusions

In this paper, multilayered PANi/PMMA/PPG-b-PEG-b-PPG@FG nanocomposites with varying PANi contents opting chemical oxidative in situ polymerization were successfully fabricated. Pre-treatment of graphite, i.e., acidic functionalization, was performed without damaging the structure and properties of filler. Subsequently, the incorporation of FG as filler led to the production of high-performance polymer/graphite nanocomposites. It was investigated by FTIR that successful fabrication of PANi/PMMA/PPG-b-PEG-b-PPG@FG nanocomposites takes place via a layer-by-layer oxidative method. The homogenous dispersion of modified filler was further evidenced by FESEM micrographs reflecting discrete interfacial bonding inside the framework of obtained nanocomposites. It was found that synthesized composites have π-π stacking and hydrogen bonding, confirming the formation of layered composites. The chemical and physical interactions resulted in improved surface area. The degradation temperature also shifted towards higher values as a result of formation of layered composites. It was revealed by DSC analysis that addition of CP from 2 to 10 wt. % results increases in T_g_, T_m_ and T_c_. This may be attributed to the enhanced chain rigidity and crosslinking. Moreover, the composites showed increased electrical conductivity with addition of PANi. The present hybrid material may be of great importance for microelectronics including polymer Li-ion batteries as well as energy-related industries in future.

## Data Availability

Data are available in the manuscript and as Appendix A.

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
