# Peer review of "Hierarchical Polyaniline Core-Shell Nanocomposites Coated on Modified Graphite for Improved Electrical Conductivity Performance"

_nanomaterials, 2022, doi:10.3390/nano12213776_

Round 1

Reviewer 1 Report

The authors' work is part of the world's extensive research on composites made of polymers and various forms of carbon. In my research, I am also looking for polymer composites with carbon nanotubes, which is why I read the work with great interest. The researchers put a lot of work into conducting the research and obtained very interesting results worth publishing.

Author Response

Dear Worthy Reviewer

Nanomaterials

Subject: Hierarchical Polyaniline Core-shell Nanocomposites Coated on Modified Graphite for Improved Electrical Conductivity Performance (Manuscript ID: nanomaterials-1962402)

Dear Sir/Miss

I am thankful to you for reviewing our manuscript and your comments encourages us to further design such a projects in this area of research.

I am highly thankful to you for your recommendation for publication of this manuscript.

Thanks and regards

Reviewer 2 Report

The paper "Hierarchical Polyaniline Core-shell Nanocomposites Coated on Modified Graphite for Improved Electrical Conductivity Performance" proposes a new approach for obtaining multifunctional PANI-PMMA nanocomposites using functional graphite as filler. Although the proposed methods are exciting and combined intelligently with already known procedures, some issues should be addressed.

1. Abstract - the paragraph from Line29 to Line 31 should be revised entirely.

2. Chemicals - specify the provenience and whether or not they were used as provided. For instance, there are a lot of Aldrich subsidiaries, so it is necessary from which you procure the chemicals.

3. Line 215- Line 216 - the "addition" is repeating.

4. Line 218 - instead of decomposition is more suitable -deconvolution!

5. Line 238 - the statement "disappearance of one peak ...: - The authors should be more precise and convincing that it is about the "grafting reaction"! Which peak? At which wavelength? Why should it be related to the grafting reaction?

6. Line 242 - the statement: " It was shown by curing characteristics ..." is not sustainable!

7. Line 332- Line 334 - "Although there is no more difference in degradation temperature .... " there is a contradiction in itself! Please clarify! Besides this, there are essential aspects of the TG that you should consider, such as the number of decomposition stages, etc. In addition, the comment from Line 334 to Line 336 is not sustained by the experimental data shown in Figure 5. Please clarify!

8. Line 361 ".. constituting higher concentration respectively." It just does not make sense; please clarify!

9. Line 369-Line 371 - the whole paragraph there should be rephrased!

10. Line 383 - instead of "disclosed," use "presented" or "shown."

Author Response

Dear Worthy Reviewer

Please find attached file containing all responses of your comments.

Thanks and regards

Reviewer 3 Report

Authors prepared  multilayered conducting composites  of PANi/PMMA/PPG-b-PEG-b-PPG@FG by in situ polymerization of aniline in the presence of functional graphite. The role of PMMA is to stabilize the growth of PANi chains through  hydrogen bonding and by providing sites for ion exchange. Various techniques, such as SEM. XRD, TG and DSC were used to characterize the obtained materials.

Some points need to be addressed in more detail:

- Introduction needs to be revised to clearly show the aim of this work;

- PANI is generally insoluble – what was thew solubility of PANI during preparation of the  nanocomposite?

- What was the molecular weight of PANI as it influences its properties;

- Scheme 1 – please correct to show hydrogen bonds between CO and HN groups;

-  Fig. 4 – please give an evidence and describe the hierarchical and core-shell structure (Title: „Hierarchical Polyaniline Core-shell Nanocomposites…”) on the basis of SEM analysis;

- Fig. 6  - please add values on the heat flow axis and indicate where T gis located;   what are the glass transitions of Poly(propylene glycol)-block-Poly(ethylene glycol)-block-poly(propylene glycol)?

- In Abstract – please correct the sentence „The fabrication of multilayered conducting blends of PANi/PMMA/PPG-b-PEG-b-PPG was carried out via in situ polymerization of aniline  monomer in the presence of functional graphite”  and „The PMMA copolymer”. Moreover,  it is  „Poly(methyl methacrylate)”.

Author Response

(The authors gave the same response as above.)

Reviewer 4 Report

The authors describe a research article entitled “Hierarchical Polyaniline Core-shell Nanocomposites Coated on Modified Graphite for Improved Electrical Conductivity Performance”. The topic of the manuscript is interesting, and the manuscript constitutes an interesting article concerning the polymer chemistry aiming at developing conducting composites. Several complementary techniques have been used to characterize the different composites. A detailed conclusion is also provided, evidencing the perspectives of this work.

The work is well-written and a well-constructed introduction has been established by the authors. Sufficient spectra and figures are included in the manuscript for comprehension and clarity. Interesting and convincing results are also presented in this work. Overall, I think that this is a manuscript that I recommend for publication after inclusion of minor revisions.

1) From my viewpoint, more details concerning the electrical measurements should be provided. At present, this section is really short.

2) What about the stability of composites over time ? Can the composites be stored for months without modifications of the electrical conductivity?

3) Do the authors observed a modification of the electrical conductivity under and without oxygen ?

Author Response

(The authors gave the same response as above.)

Round 2

Reviewer 3 Report

Authors provided proper replies to reviewer's comments, and the revised manuscript can be published.